# Additive Cytotoxic and Colony-Formation Inhibitory Effects of Aspirin and Metformin on *PI3KCA*-Mutant Colorectal Cancer Cells

**DOI:** 10.3390/ijms25105381

**Published:** 2024-05-15

**Authors:** Joana Gonçalves, Sara Pinto, Francisca Carmo, Cláudia Silva, Nelson Andrade, Fátima Martel

**Affiliations:** 1Unit of Biochemistry, Department of Biomedicine, Faculty of Medicine, University of Porto, 4200-319 Porto, Portugal; up20207713@edu.fc.up.pt (J.G.); up201905133@edu.fc.up.pt (S.P.); up201606420@edu.icbas.up.pt (F.C.); nandrade@med.up.pt (N.A.); 2LAQV/REQUIMTE, Department of Chemical Sciences, Faculty of Pharmacy, University of Porto, 4050-453 Porto, Portugal; claudiasilva@med.up.pt; 3Instituto de Investigação e Inovação em Saúde (i3S), University of Porto, 4200-135 Porto, Portugal

**Keywords:** metformin, aspirin, HT-29, Caco-2, PI3K, anticancer effect

## Abstract

Human malignancies are one of the major health-related issues throughout the world and are anticipated to rise in the future. Despite huge investments made in anticancer drug development, limited success has been obtained and the average number of FDA approvals per year is declining. So, an increasing interest in drug repurposing exists. Metformin (MET) and aspirin (ASP) possess anticancer properties. This work aims to test the effect of these two drugs in combination on colorectal cancer (CRC) cells in vitro. The effects of MET and/or ASP on cell proliferation, viability, migratory ability, anchorage-independent growth ability (colony formation), and nutrient uptake were determined in two (HT-29 and Caco-2) human CRC cell lines. Individually, MET and ASP possessed antiproliferative, cytotoxic, and antimigratory effects and reduced colony formation in HT-29 cells (*BRAF*- and phosphatidylinositol-4,5-bisphosphate 3-kinase catalytic subunit α (*PI3KCA*)-mutant), although MET did not affect either ^3^H-deoxy-D-glucose or ^14^C-butyrate uptake and lactate production, and ASP caused only a small decrease in ^14^C-butyrate uptake. Moreover, in these cells, the combination of MET and ASP resulted in a tendency to an increase in the cytotoxic effect and in a potentiation of the inhibitory effect on colony formation, although no additive antiproliferative and antimigratory effects, and no effect on nutrient uptake and lactate production were observed. In contrast, MET and ASP, both individually and in combination, were almost devoid of effects on Caco-2 cells (*BRAF*- and *PI3KCA*-wild type). We suggest that inhibition of PI3K is the common mechanism involved in the anti-CRC effect of both MET, ASP and their combination and, therefore, that the combination of MET + ASP may especially benefit *PI3KCA*-mutant CRC cases, which currently have a poor prognostic.

## 1. Introduction

Cancer is a major health concern and is anticipated to be the primary cause of death in all countries in the 21st century [1,2]. Among cancer types, colorectal cancer (CRC) is the third cause of cancer-related death worldwide and the second cause in Europe. Despite advances in surgical and adjuvant treatment over the past two decades, CRC is often incurable in advanced stages with current treatments. For this reason, the incidence and mortality of CRC continue to increase worldwide, contributing significantly to the global increase in cancer mortality [1,2].

The process of de novo development of new anticancer drugs is expensive, time-consuming, and presents a high risk of drug failure. Moreover, chemotherapeutic agents frequently present significant toxicity and badly impact the quality of life of cancer patients. This has prompted the pharmaceutical industry to seek alternative strategies that may facilitate and accelerate this whole process. In particular, the drug repurposing strategy, also known as the repositioning or reprofiling strategy—a novel concept of using existing approved drugs for different disorders than their initial purpose—is a potential source of new treatment options for cancer patients. Drug repositioning is expected to save time, energy, and money and allow direct entry of drugs into clinical trials as they have already gone through toxicity and safety profiling [3,4,5].

Aspirin (ASP) and metformin (MET) have a wide and diverse spectrum of pharmacological activities. ASP is well known for its anti-inflammatory potential resulting from the inhibition of cyclooxygenase-1 (COX-1) and COX-2 [6], whereas MET is used as the first-line treatment for type 2 diabetes mellitus (T2DM) [7]. More recently, ASP and MET have been also considered for their potential anticancer activities [3,4,5]. Specifically in relation to CRC cancer, MET has been shown to reduce CRC incidence, being more effective in diabetic patients [8,9,10,11,12,13]. Similarly, a low dose of ASP has been associated with a decrease in CRC occurrence, recurrence, and mortality [14,15,16], and ASP has been shown to be well tolerated by patients in adjuvant anticancer therapies [17]. Accordingly, several in vitro and in vivo preclinical studies reported that ASP and MET possess anticancer effects in CRC cells [18,19,20,21].

Because CRC incidence is higher in T2DM patients [22,23] and MET, the first-choice drug in these patients [7], also presents CRC antagonizing effects, there is growing interest in evaluating the possible additional benefit in relation to CRC risk and/or therapy of low dose ASP in MET-using T2DM patients. In this context, the effects of this combination in relation to CRC risk and survival were analyzed in some retrospective cohort studies [24,25,26,27], and clinical trials to assess the potential beneficial effect of ASP + MET combinations in CRC patients are currently ongoing [8,28,29]. However, the effect of combining MET and ASP has not yet been properly investigated in in vitro CRC models. Indeed, only two studies have previously evaluated the in vitro effect of this combination in CRC cells, with limited information on the anticancer effect of this combination being obtained [30,31]. So, it is decidedly important to further investigate the effect of the ASP + MET combination in vitro, by using more CRC cell lines and by simultaneously evaluating several cancer-related parameters.

For this reason, we decided to evaluate the effects of MET and ASP, alone or in combination, on cell viability, proliferation, migration, colony-forming ability, and metabolic features of two CRC cell lines with distinct characteristics (HT-29 and Caco-2). These two cell lines were chosen because they are CRC cell lines with distinct cellular (grades of differentiation [32] and migratory and metastatic potential [33,34]) and molecular phenotypic (Caco-2 cells are *BRAF*- and *PI3KCA* (phosphatidylinositol-4,5-bisphosphate 3-kinase catalytic subunit α)-wild type, whereas HT-29 cells are *BRAF*- and *PI3KCA*-mutant) characteristics. Moreover, these two cell lines also differ in CpG island methylator phenotype (CIMP) [35,36].

## 2. Results

### 2.1. Effect of Metformin (MET) and/or Aspirin (ASP) on HT-29 Cellular Proliferation

As shown in Figure 1a, MET and ASP concentration-dependently reduced HT-29 cell proliferation. However, the combination of these two drugs did not result in an increased antiproliferative effect (Figure 1b).

### 2.2. Effect of Metformin (MET) and/or Aspirin (ASP) on HT-29 Cell Viability

The effect of MET and/or ASP on HT-29 cell viability was assessed through three distinct methods, based on different principles for the evaluation of cytotoxicity (Figure 2). Both MET and ASP, at the tested concentrations, were found to be cytotoxic with the sulforhodamine B (SRB) and lactate dehydrogenase (LDH) assays (Figure 2a,e), although this effect was not observed with the resazurin reduction assay (Figure 2c). The cytotoxic effect was more consistently observed with ASP than with MET (Figure 2). When these drugs were combined, their cytotoxicity tended to be higher than when they were tested individually (Figure 2b,d,f).

### 2.3. Effect of Metformin (MET) and/or Aspirin (ASP) on HT-29 Cell Migration Rates

The effect of MET and/or ASP on the migratory ability of HT-29 cells was next evaluated. Only the highest concentration of MET and ASP (1 mM) was able to reduce the migratory capacity of the cells (Figure 3a). The intermediate concentrations of both compounds (0.1 mM and 0.5 mM, respectively), which showed no antimigratory effect individually, were also devoid of effect when combined, although a tendency was observed (Figure 3b).

### 2.4. Effect of Metformin (MET) and/or Aspirin (ASP) on Anchorage-Ndependent Growth Ability

Interestingly, both MET (0.1 mM) and ASP (0.5 mM) reduced the anchorage-independent growth (colony formation) ability of HT-29 cells, and this inhibitory effect was more profound when both compounds were combined (Figure 4).

### 2.5. Effect of Metformin (MET) and/or Aspirin (ASP) on HT-29 Nutrient Uptake and Metabolism

The effect of combining MET and ASP on the metabolic characteristics of HT-29 cells was also evaluated. As shown in Figure 5, apart from an inhibitory effect of ASP on ^14^C-butyrate (^14^C-BT) uptake, MET, ASP, and their combination were devoid of effect on ^3^H-deoxy-d-glucose (^3^H-DG) and ^14^C-BT uptake and lactate production.

### 2.6. Effect of Metformin (MET) and/or Aspirin (ASP) on Caco-2 Cell Proliferation, Viability, Migration Rates, Anchorage-Independent Growth Ability, and Nutrient Uptake

In the Caco-2 cell line, neither MET, ASP nor their combination caused any significant alteration in cell proliferation rates, viability, migratory rates, and ^3^H-DG uptake (Figure 6a–d,g). The only significant effects were a decrease in anchorage-independent growth ability and in ^14^C-BT uptake, observed in the MET + ASP group (Figure 6e,f,h).

## 3. Discussion

The repurposing strategy is a potential source of new treatment options for cancer patients, that is expected to save time, energy, and money and to allow direct entry to clinical trials as the drugs have already gone through toxicity and safety profiling [3,4,5]. The best repurposable oncological drug candidates are agents whose original patent protection has already expired, and for which it is possible to create a formulation enabling, together with a new therapeutic indication, new patent protection [4]. The antidiabetic MET and the non-steroidal anti-inflammatory drug ASP, both of which have shown anticancer properties, fulfill this criterion.

Because CRC incidence is higher in T2DM patients [24,25] and MET, the first-choice drug in these patients [7], also presents CRC antagonizing effects, there is growing interest in evaluating if there is an additional benefit in relation to CRC risk and/or therapy of a low-dose ASP in MET-using T2DM patients. So, clinical trials to test the potential beneficial effect of ASP + MET combinations in CRC patients are currently undergoing [8,28,29]. However, the effects of this combination in relation to CRC risk and survival analyzed in retrospective cohort studies have been inconclusive [24,25,26,27], and only two previous studies evaluated the in vitro effect of this combination in CRC cells, with limited information on the anticancer effect of this combination being obtained [30,31]. So, more in vitro studies are urgently needed because the effects of combining MET and ASP have not yet been properly investigated in in vitro CRC models. In this context, this study evaluated, for the first time, the effects of MET and ASP, alone and in combination, on cell viability, proliferation, migration, colony-forming ability, and metabolic features of two CRC cell lines with distinct characteristics (HT-29 and Caco-2).

We verified that, individually, both MET and ASP possessed antiproliferative, cytotoxic, and antimigratory effects, and inhibited anchorage-independent colony formation of HT-29 cells. These results agree with previous studies. Namely, MET was previously shown to have antiproliferative and proapoptotic effects in mice CRC [18,20,37] and to inhibit the proliferation of the HT-29 cell line [20]. Similarly, ASP was previously reported to decrease proliferation and to alter cell cycle distribution in the HT-29 cell line [21], to decrease cell viability and increase apoptosis in six CRC cell lines (HRT-18, SW480, HT-29, DLD-1, LoVo, and HCT116 cells) [38], and to reduce proliferation and to increase apoptosis and autophagy in three CRC cell lines (RKO, SW480, and HCT116 cells) [30].

When the effect of combining MET and ASP was tested in HT-29 cells, we verified that the antiproliferative and antimigratory effects of the individual drugs did not increase. In contrast, the cytotoxic effect tended to increase and a potentiation of the inhibitory effect on colony formation ability was found.

The lack of consistent additive/synergic effects of MET + ASP in HT-29 cells agrees with previous observations in other in vitro CRC preclinical models. Indeed, although it was verified, using three distinct CRC cell lines (RKO, SW480, and HCT116 cell lines), that the combination of MET + ASP has a striking additive effect on AMPK (5′adenosine monophosphate-activated protein kinase) activation and mTOR (mammalian target of rapamycin) inhibition, with increased autophagy [30], and that this combination showed an additive/synergic effect in tumor spheroids generated in suspension from patient-derived CRCs, there was no additive effect when these two compounds were combined in 2D cultures of the CRC HCT-15 cell line [31]. Moreover, a lack of consistent effects of MET + ASP was also observed in other preclinical cancer models. In fact, although an additive/synergic in vitro effect of MET and ASP combination in inhibiting breast, lung, pancreatic, prostate, and thyroid cancer and hepatocellular carcinoma cell growth and increasing cell death were described [39,40,41,42,43], a lack of additive/synergic effect in breast cancer cell lines was also described [42].

Moreover, the observation of an additive anticancer effect of MET and ASP in relation to only some of the CRC-related cellular characteristics of HT-29 cells may explain retrospective cohort studies, in which variable effects of this combination were reported: either no additional benefit of this combination in overall cancer [44] and CRC [26] risk, a reduction in CRC risk [24] and in 5-year [25] survival of CRC patients [27], or an increase in cancer-related mortality [44].

In contrast, consistent additive/synergic effects of MET or ASP with other compounds in relation to CRC have been found (namely, MET with rapamycin (sirolimus) [45], thymoquinone [46] or oxaliplatin [47] and ASP with platinum derivatives [19], selenium [48] and an anti-PD-L1 antibody [49]).

We also evaluated the effect of the MET and ASP combination in relation to nutrient cellular uptake and metabolism by HT-29 cells, by focusing on BT and glucose. The short-chain fatty acid BT, one of the main end products of anaerobic bacterial fermentation of dietary fiber in the human colon, is the main energy source of normal colonic epithelial cells and has important homeostatic functions at this level, including the ability to prevent/inhibit carcinogenesis [50]. However, CRC epithelial cells show a reduction in BT cellular uptake associated with an increase in the rates of glucose uptake, which becomes their primary energy source [50]. This increase in glycolysis rates with lactate production is a feature—known as the Warburg effect—characteristic of several cancer types [32,51]. We verified that, apart from a small reduction in ^14^C-BT uptake caused by ASP, none of the drugs, either alone or in combination, affected ^3^H-DG and ^14^C-BT cellular uptake and lactate production by HT-29 cells. We can thus conclude that their anticancer effects in HT-29 cells are not associated with a decrease in glucose or butyrate cellular uptake.

The lack of effect of MET and ASP combination on ^14^C-BT uptake, despite a small inhibition of this parameter with ASP alone, indicates that metformin prevents ASP-induced inhibition of ^14^C-BT uptake. Because MET can affect other membrane transporters (e.g., glucose transporters) [52], we hypothesize that this effect is the result of MET interaction with BT transporters.

Very distinctly from the effects observed in HT-29 cells, MET, ASP, and their combination were almost devoid of effect (apart from inhibition of anchorage-independent growth and ^14^C-BT uptake found with MET + ASP) in another CRC cell line, the Caco-2 cell line.

Although both cell lines were isolated from human adenocarcinomas, HT-29 and Caco-2 cells possess varying grades of differentiation [32] and different migratory and metastatic potentials [33,53]. For instance, HT-29 cells are invasive and metastatic in vivo, but not in vitro, whereas Caco-2 cells are noninvasive [33], and HT-29 cells, but not Caco-2 cells, have the ability to grow as cell-forming spheres [53]. Consistent with these phenotypic differences, molecular phenotyping of these cell lines also shows some differences. Indeed, although both cell lines show microsatellite-stable (MSS) and chromosomal instability pathway (CIN) phenotypes and harbor mutations in the tumor-suppressing protein p53 but not in the oncogenes KRAS and PTEN, Caco-2 cells do not carry a mutation in *BRAF* and in *PI3KCA* (phosphatidylinositol-4,5-bisphosphate 3-kinase catalytic subunit α) genes, whereas HT-29 cells carry a V600E mutation in the *BRAF* oncogene and a P449T mutation in PI3KCA. Moreover, these two cell lines also differ in CpG island methylator phenotype (CIMP) [35,36].

*PIK3CA* is a frequently mutated oncogene in CRCs, with mutations in *PIK3CA* being observed in 20–25% of CRCs [54]. The *PIK3CA* gene encodes the p110a catalytic subunit of PI3 kinase (PI3K), whose signaling pathway plays an important role in human carcinogenesis [55,56]. *PIK3CA*-activating mutations lead to hyperactivation of the pro-tumorigenic PI3K/AKT pathway [56]. PI3K is also a downstream effector of EGFR signaling, and thus mutations in *PI3KCA* may also affect anti-EGFR therapy responsiveness. Accordingly, *PI3KCA* mutations are predictive of worse outcomes with anti-EGFR therapy [57].

Potential mechanisms for the CRC anticancer effect of MET include activation of AMPK/inhibition of mTOR, inhibition of the PI3K, and mitogen-activated protein kinase (MAPK) signaling pathways, lowering of hyperinsulinemia, inhibition of cellular respiration through inhibition of respiratory complex I, and modulation of inflammatory response [3,4,5,14,58,59]. As to ASP, potential mechanisms involved in its CRC anticancer effect include activation of AMPK/inhibition of mTOR signaling and inhibition of inflammatory components such as COX-1/COX-2 and nuclear factor kappa B (NFκB) [3,4,5,14,29]). Importantly, one of the downstream consequences of *PIK3CA* mutations in CRC is COX-2 upregulation [60]. ASP may therefore suppress CRC cancer cell growth and induce apoptosis by blocking the PI3K pathway through COX-2 inhibition, and this effect will differ according to PIK3CA mutational status [61,62,63]. In agreement with this, PIK3CA mutations are associated with the benefit of ASP for secondary CRC prevention [57].

As stated above, *PI3KCA* plays an important role in carcinogenesis, and it is a common mechanism involved in the anti-CRC effect of both MET and ASP. The observation that the effect of MET, ASP, and their combination is very distinct in HT-29 cells (which possess mutated *PI3KCA*) and Caco-2 cells (which possess wild-type *PI3KCA*) led us to hypothesize that inhibition of the PI3K pathway may be the common mechanism involved in the anti-CRC effect of both these drugs. This conclusion explains why these agents were much more effective in HT-29 cells (which have a hyperactivated PI3K pathway) than in Caco-2 cells, and why their combination did not result in a consistent additive effect in the HT-29 cell line.

There may be some possible limitations in this study. First, only two CRC cell lines were used. Second, the results observed in the present in vitro study should be confirmed in vivo. Finally, the concentrations of MET and ASP used are above the current therapeutic blood levels of these drugs in humans. Nevertheless, targeted drug delivery, using e.g., nanocarriers, may overcome this problem [64].

In conclusion, MET and ASP were found to have anti-tumoral effects in the HT-29 cell line (*BRAF*- and *PI3KCA*-mutant), but not in the Caco-2 cell line (*BRAF*- and *PI3KCA*-wild type). Moreover, in HT-29 cells, the combination of MET and ASP resulted in a tendency to an increase in the cytotoxic effect and in a potentiation of the inhibitory effect on colony formation ability, although no additive antiproliferative and antimigratory effects, and no effect on nutrient uptake and lactate production were observed. We hypothesize that inhibition of PI3K is the common mechanism involved in the anti-CRC effect of both MET, ASP, and their combination and that, therefore, the combination of MET + ASP will probably benefit *PI3KCA*-mutant CRC cases, which have currently a poor prognostic (as already observed in relation to ASP only; see above).

## 4. Materials and Methods

### 4.1. Materials

ASP (Sigma, St. Louis, MO, USA); MET (European Pharmacopeia, Strasbourg, France); ^3^H-Deoxy-D-glucose (2-[1,2-^3^H(N)]-deoxy-D-glucose: specific activity 60 Ci/mmol); ^14^C-n-butyric acid ([1-^14^C-n-butyric acid]; specific activity 55 mCi/mmol) (American Radiolabeled Chemicals Inc., St Louis, MO, USA); ^3^H-thymidine ([methyl-^3^H]-thymidine; specific activity 79 Ci/mmol) (GE Healthcare GmbH, Freiburg, Germany); trypsin-EDTA 0.25% (PAN-Biotech™, Aidenbach, Germany); crystal violet, triton X-100, trichloroacetic acid (TCA) (Merck, Darmstadt, Germany); Dulbecco’s Modified Eagle Medium (DMEM; catalogue #D5796), Minimum Essential Medium (MEM; catalogue #M-0643), antibiotic/antimycotic solution (100 U/mL penicillin, 0.1 mg/mL streptomycin, and 0.25 µg/mL amphotericin B), bovine serum albumin (BSA), fetal bovine serum (FBS), reduced nicotinamide adenine dinucleotide (NADH), resazurin, hydroxyethylpiperazine-N0-2-ethanesulfonic acid (HEPES), sodium pyruvate; sulforhodamine B (SRB), tris-HCl (tris(hydroxymethyl)-aminomethane hydrochloride), trypsin–ethylenediaminetetraacetic acid (EDTA) solution (Sigma, St. Louis, MO, USA).

### 4.2. Cell Culture

Two colorectal adenocarcinoma cell lines were used: HT-29 (ATCC HTB-38; passage numbers 7–30) and Caco-2 (ATCC HTB-37; passage numbers 31–32).

HT-29 cells were cultured in high glucose (4.5 g/L) DMEM medium supplemented with 10% (*v*/*v*) heat-inactivated fetal bovine serum (FBS) and 1% (*v*/*v*) antibiotic/antimycotic solution and Caco-2 cells were cultured in MEM medium supplemented with 15% heat-inactivated FBS and 1% antibiotic/antimycotic solution. Cells were maintained in a humidified atmosphere of 95% air and 5% CO_2_. The culture medium was changed every 3–4 days, and the culture was split every 7 days. For sub-culturing, the cells were removed enzymatically (0.25% trypsin–EDTA, 4 min, 37 °C), split 1:4–1:5 (HT-29 cells) or 1:3–1:4 (Caco-2 cells), and sub-cultured in plastic culture dishes (21 cm^2^; diameter 60 mm; TPP^®^, Trasadingen, Switzerland).

For determination of cell viability, proliferation rates, migration rates, ^3^H-DG and ^14^C-BT uptake, and glucose metabolism, cells were seeded on 24-well plastic cell culture dishes (2 cm^2^ Ø 16 mm; TPP^®^) and used at 80–90% confluency (5–6 days-old culture for HT-29; 6–11 days-old for Caco-2 cells). For the resazurin assay, cells were seeded in 96-well plates (TPP^®^). For determination of anchorage-independent growth ability, the cells were seeded in 6-well plates and used after 19 days.

### 4.3. Treatment of the Cells

The concentrations of MET (10 μM, 100 μM, and 1 mM) and ASP (100 µM, 500 µM, and 1 mM) to test were chosen based on previous works of our group [65,66]. The cells were treated for 24 h with MET and/or ASP (or the respective solvents) in an FBS-free culture medium. MET was dissolved in water and ASP in ethanol (100%). The solutions containing these drugs were added to the culture medium in a final volume of 1% (*v*/*v*). For controls, 1% (*v*/*v*) of the solvent of each compound was added to the culture medium.

### 4.4. Determination of Cell Proliferation Rates

#### Incorporation of ^3^H-Thymidine Assay

Cellular proliferation rates were determined through the incorporation of ^3^H-thymidine, which quantifies DNA synthesis [67]. Cells were exposed for 24 h to MET and/or ASP (or the respective solvents) and incubation with ^3^H-thymidine 0.125 μCi/mL was carried out in the last 5 h of treatment. The excess of ^3^H-thymidine was rinsed off with 10% TCA, followed by a 20 min incubation with NaOH 1 M (280 μL/well). Before the addition of scintillation fluid, the lysate was neutralized with 5 mM HCl. Radioactivity was then quantified using liquid scintillometry (LKB Wallac 1209 Rackbeta, Turku, Finland). The cellular DNA synthesis rate was expressed as the incorporation of ^3^H-thymidine (µCi/mg protein).

### 4.5. Determination of Cell Viability

Three distinct methods, of evaluating cytotoxicity through distinct principles, were used. The sulforhodamine B (SRB) assay quantifies total cell proteins, the resazurin reduction assay detects metabolically active cells, and the lactate dehydrogenase (LDH) activity assay measures the activity of extracellular LDH, which is released from cells with a damaged cell membrane.

#### 4.5.1. SRB Assay

Under mildly acidic conditions, SRB binds to protein basic amino acid residues in TCA-fixed cells to provide a sensitive index of cellular protein amount [68]. After exposure of the cells for 24 h to MET and/or ASP (or the respective solvents), the medium was replaced with 500 μL PBS 1×, and 62.5 μL of ice-cold 50% (*w*/*v*) TCA was added to each well to fix cells, for 1 h at 4 °C in the dark. Then, cells were washed with tap water to remove excess TCA, air dried, and then stained for 15 min with 125 μL 0.4% (*w*/*v*) SRB dissolved in 1% acetic acid per well. After SRB removal, cells were rinsed with 1% acetic acid to remove residual die, and then air dried again. The bound die was solubilized with 375 μL of 10 mM Tris-NaOH solution (pH = 10.5). To obtain absorbance (540 nm) values below 0.7, the contents of each well were diluted and placed in a 96-well plaque where the absorbance was measured using a microplate reader (Thermo Fisher Scientific, Waltham, MA, USA).

#### 4.5.2. Resazurin Reduction Assay

This assay estimates the number of viable cells by measuring the reduction of resazurin into resorufin [69]. After treatment with MET and/or ASP (or the respective solvents), resazurin solution (0.1 mg/mL) (10 µL) was added to each well of the 96-well plate and incubated for 3 h at 37 °C and 5% CO_2_. The fluorescence intensity was then measured at a 530 nm excitation wavelength and a 590 nm emission wavelength using a SpectraMAX Gemini EM fluorescent microplate reader (Molecular Devices, Silicon Valley, CA, USA).

#### 4.5.3. Extracellular LDH Activity Assay

Cell death is correlated to leakage of LDH to the extracellular medium and measurement of LDH activity, through the quantification of the decrease in absorbance of NADH during the reduction of pyruvate to lactate, can be used to evaluate cell viability [70]. After exposure of the cells for 24 h to MET and/or ASP (or the respective solvents), the extracellular medium was removed and 50 μL of the extracellular medium was added to 250 µL of a mixture of phosphate/pyruvate (50 mM/0.63 mM) solution and NADH (11.3 mM) solution. Absorbance values were measured in a 96-well plaque at 340 nm, and the decrease in absorbance value, in this time interval, was calculated. LDH activity was expressed as the percentage of extracellular activity in relation to total cellular LDH activity, which represents 100% cell death. To determine the total activity, control cells were solubilized with 500 μL of 0.1% (*v*/*v*) Triton X-100 and incubated for 1 h at 4 °C, and then the sample was processed in the same way as the others.

### 4.6. Determination of Cell Migration Rates

#### Scratch Injury Assay

Briefly, cell monolayers were scratched with a 10 μL pipette tip and were afterward treated for 24 h with MET and/or ASP (or the respective solvents) [71]. Images were obtained at 0 and 24 h after the scratch, and quantification was performed using the ImageJ ImageJ v1.53c software (NIH, Bethesda, MD, USA).

### 4.7. Determination of Anchorage-Independent Growth Ability

#### Soft Agar Colony Formation Assay

This assay measures anchorage-independent colony growth [72]. Briefly, 6-well plates were coated with 0.6% noble agar in complete media (1.5 mL/well) and allowed to solidify at room temperature for 30 min. After exposure of the cells for 24 h to MET and/or ASP (or the respective solvents), 15 × 10^3^ cells were mixed with complete media containing 0.3% noble agar, which was added to the top of the base agar, and allowed to solidify for 30 min. Complete media (200 μL) was then added to each well to prevent drying, and the wells were incubated for 19 days at 37 °C under 5% CO_2_. Afterward, colonies were stained with 0.01% crystal violet for 1 h, and the images were captured with a microscope. Only colonies with diameter > 50 μM were counted. The average number of colonies was quantified using the ImageJ software (NIH, Bethesda, MD, USA).

### 4.8. Determination of ^3^H-Deoxy-d-glucose (^3^H-DG) and ^3^H-Butyrate (^14^C-BT) Uptake

Cells were exposed for 24 h to MET and/or ASP (or the respective solvents) and ^3^H-DG or ^14^C-BT uptake was afterward evaluated. The experiments were performed using GF-HBS buffer (glucose-free HEPES buffered saline) (composition, in mM: 140 NaCl, 5 KCl, 20 HEPES, 2.5 MgSO_4_, 1 CaCl_2_; pH 7.4) at 37 °C.

The culture medium was removed, and the cells were washed with 300 μL GF-HBS buffer. Then, cell monolayers were preincubated for 20 min in GF-HBS buffer followed by an incubation with 10 nM ^3^H-DG or 10 µM ^14^C-BT for 6 min or 3 min, respectively. When tested, MET and/or ASP (or the respective solvents) were also present during both pre-incubation and incubation periods. Incubation was stopped by rapidly removing the incubation medium, placing the cells on ice, and rinsing the cells with 500 μL ice-cold GF-HBS buffer. The cells were then solubilized with 300 μL 0.1% (*v*/*v*) Triton X-100 (in 5 mM Tris-HCl, pH 7.4), and placed at 4 °C overnight. The radioactivity of the samples was quantified using liquid scintillometry (LKB Wallac Liquid Scintillation Counter 1209, Turku, Finland).

### 4.9. Determination of Glucose Metabolism

#### Quantification of Lactate Metabolism

Cells were exposed for 24 h to MET and/or ASP (or the respective solvents). Then, the extracellular medium was collected and centrifuged at 8000× *g* for 10 min. Lactate concentration was then measured via the lactate oxidase/peroxidase colorimetric assay, as indicated by the manufacturer (Olympus Life and Material Science Europa GmbH, Hamburg, Germany).

### 4.10. Protein Determination

The protein content of the cell monolayers was determined as described by Bradford [73], using human serum albumin as standard.

### 4.11. Statistics

Arithmetic means are given with standard error of the mean (SEM). The value of n indicates the number of replicates; the number of assays indicates the number of distinct experiments. Statistical significance of the difference between three or more groups was evaluated by a one-way ANOVA test, followed by the Student–Newman–Keuls test. For comparison between the two groups, the Student’s *t*-test was used. Differences were considered significant when *p* < 0.05. Analyses were carried out using the GraphPad Prism version 9.0 software (San Diego, CA, USA).

## Figures and Tables

**Figure 1 ijms-25-05381-f001:**
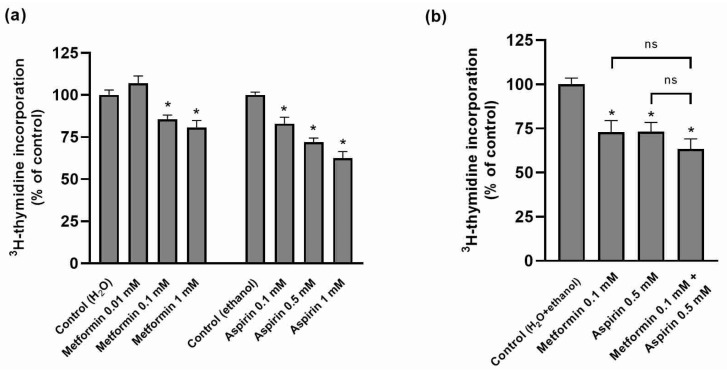
Effect of metformin and/or aspirin on the proliferation of HT-29 cells. (**a**) Effect of increasing concentrations of metformin and aspirin (n = 4–6; 3 assays); (**b**) effect of the combination of metformin and aspirin (n = 17; 5 assays). Results are expressed as arithmetic means ± SEM. * significantly different from control; ^ns^ not significantly different from each other.

**Figure 2 ijms-25-05381-f002:**
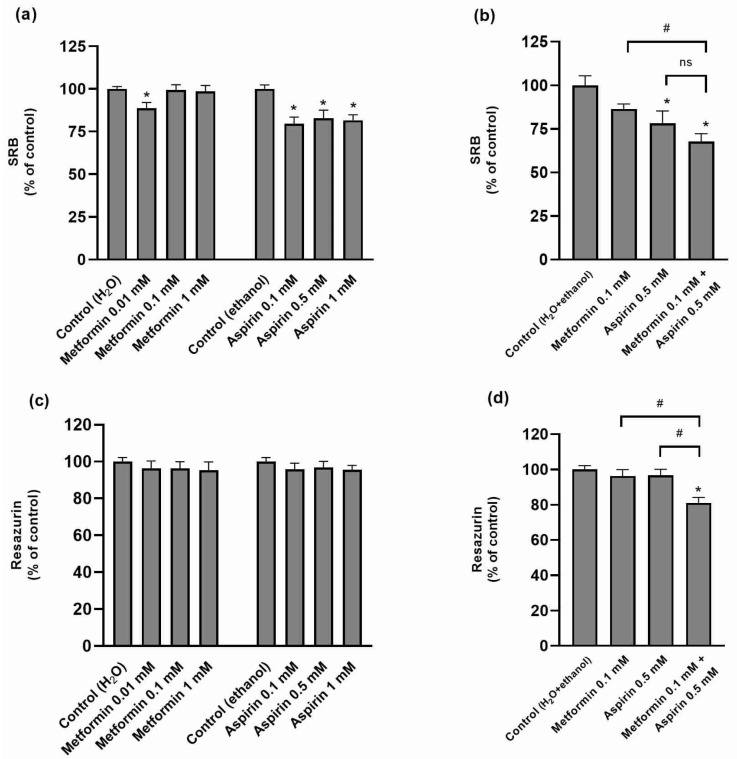
Effect of metformin and/or aspirin on the viability of HT-29 cells, evaluated with the sulforhodamine B (SRB), resazurin reduction, and lactate dehydrogenase (LDH) assays. (**a**,**b**) Cell viability assessed with the SRB assay (n = 9–20; 3–5 assays); (**c**,**d**) cell viability assessed with the resazurin assay (n = 15; 3 assays); (**e**,**f**) cell viability assessed with the LDH assay (n = 9–13; 3–4 assays). (**a**,**c**,**e**) Effect of increasing concentrations of metformin and aspirin; (**b**,**d**,**f**) effect of the combination of metformin and aspirin. Results are expressed as arithmetic means ± SEM. * significantly different from control; ^#^ significantly different from each other; ^ns^ not significantly different from each other.

**Figure 3 ijms-25-05381-f003:**
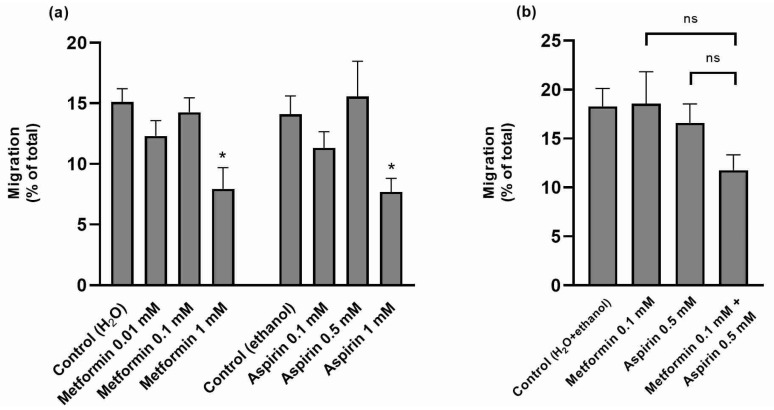
Effect of metformin and/or aspirin on the migration of HT-29 cells. (**a**) Effect of increasing concentrations of metformin and aspirin (n = 9; 3 assays); (**b**) Effect of the combination of metformin and aspirin (n = 6; 2 assays). Results are expressed as arithmetic means ± SEM. * significantly different from control; ^ns^ not significantly different from each other.

**Figure 4 ijms-25-05381-f004:**
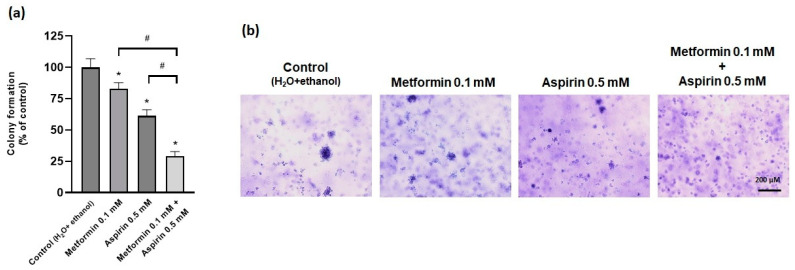
Effect of metformin and/or aspirin on anchorage-independent growth ability of HT-29 cells. (**a**) Results are expressed as arithmetic means ± SEM (n = 6; 2 assays). * significantly different from control; ^#^ significantly different from each other. (**b**) representative images of the colonies. Scale bar: 200 µm.

**Figure 5 ijms-25-05381-f005:**
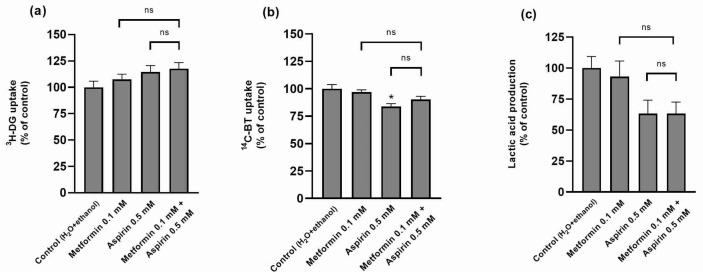
Effect of metformin and/or aspirin on nutrient uptake and metabolism in HT-29 cells. (**a**) Effect of the combination of metformin and aspirin on ^3^H-deoxy-d-glucose (^3^H-DG) uptake; (**b**) effect of the combination of metformin and aspirin on ^14^C-butyrate (^14^C-BT) uptake; (**c**) effect of the combination of metformin and aspirin on lactate production. Results are expressed as arithmetic means ± SEM (n = 12; 3 assays). * significantly different from control; ^ns^ not significantly different from each other.

**Figure 6 ijms-25-05381-f006:**
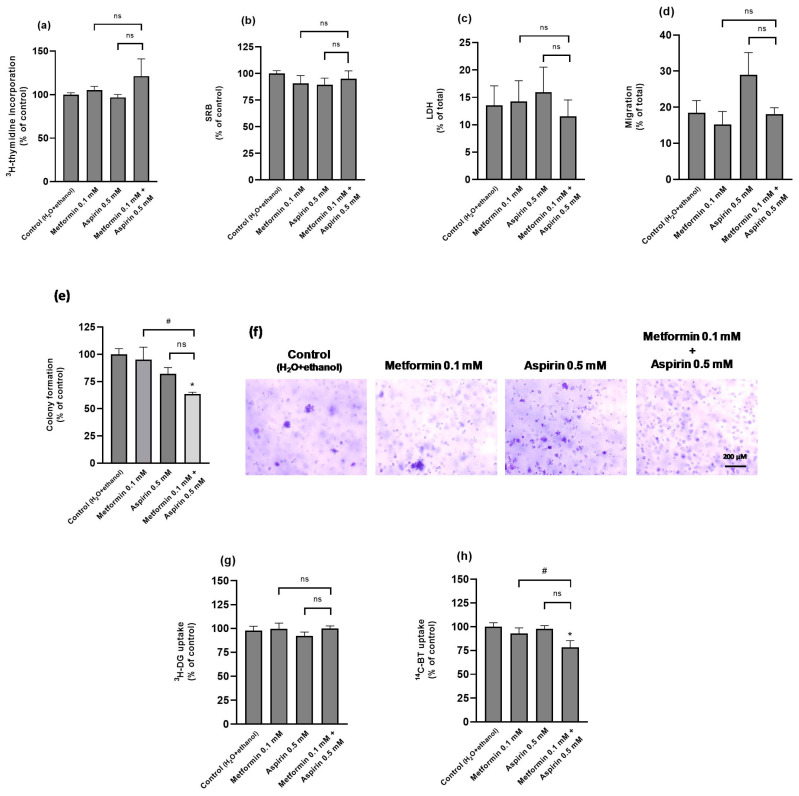
Effect of metformin and/or aspirin on Caco-2 cells. (**a**) Effect on cell proliferation (n = 6; 2 assays); (**b**,**c**) effect on cell viability, evaluated with the (**b**) SRB and (**c**) LDH assays (n = 7–8; 2 assays); (**d**) effect on cell migration (n = 5–6; 2 assays); (**e**,**f**) effect on anchorage-independent growth ability (n = 4; 2 assays), showing results expressed as arithmetic means ± SEM (**e**), and representative images of the colonies (**f**); (**g**) effect on ^3^H-deoxy-d-glucose (^3^H-DG) uptake (n = 8; 2 assays); and (**h**) effect on ^14^C-butyrate (^14^C-BT) uptake (n = 8; 2 assays). Results are expressed as arithmetic means ± SEM. * significantly different from control; ^#^ significantly different from each other; ^ns^ not significantly different from each other.

## Data Availability

The data presented in this study are available on request from the corresponding author.

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
