# Peer review of "Additive Cytotoxic and Colony-Formation Inhibitory Effects of Aspirin and Metformin on PI3KCA-Mutant Colorectal Cancer Cells"

_ijms, 2024, doi:10.3390/ijms25105381_

Round 1

Reviewer 1 Report (Previous Reviewer 1)

Comments and Suggestions for Authors

I have no further revisions to suggest.

Comments on the Quality of English Language

I think that the English language is good, but check the manuscript for some minor errors.

Author Response

Thank you for your revision.

Reviewer 2 Report (Previous Reviewer 3)

Comments and Suggestions for Authors

Authors made the related correction for some comments that I cited in my last review report, however I still don’t see the answer of the two following questions that I asked:

1-     Analysis: Authors did not state what is the positive/ negative control that they used in their test.

2-     In terms of novelty: The paper subject is like some previous papers in the literature, e.i: https://doi.org/10.1186/s12885-020-07564-z, that’s why the authors need to highlight what is the difference between their study and the previous studies in the literature, saying e.i: it is the first study of …..

I would suggest getting these answers separately or highlighted in RED in the manuscript before proceeding with the paper acceptance in the present form.

Comments on the Quality of English Language

No comments

Author Response

Thank you for the revision. We next reply to your comments.

1- Analysis: Authors did not state what is the positive/ negative control that they used in their test.

Answer: I confess I do not understand what is the information lacking. In the Methods section, we mentioned how the controls were made (the controls contain only the same volume of solvent of the drugs to be tested). Anyway, we changed the sentence in order to make it clearer (lines 362-364). Moreover, we added the information about the solvent used in each control to all the Figures. I hope this is the information you asked.

2-     In terms of novelty: The paper subject is like some previous papers in the literature, e.i: https://doi.org/10.1186/s12885-020-07564-z, that’s why the authors need to highlight what is the difference between their study and the previous studies in the literature, saying e.i: it is the first study of …..

Answer: The paper you mention (Higurashi et al. 2020, which was included in our reference list (ref 29)) is a clinical trial, and our study is an in vitro study with two CRC cell lines. So, although the subject is the same (combination of metformin+aspirin in CRC) in both works, they are quite distinct, and the information of these works give is distinct (in vitro vs. clinical trial) and complementary. Importantly, only two previous in vitro studies with CRC cells and the combination metformin+aspirin were done before the present work, and only one of these used CRC cell lines, as the other used patient-derived CRC cells. Moreover, these two studies were rather limited as they only evaluated autophagy (in one study) and tumor spheroid formation (in the other). Importantly, the effect of this combination in relation to CRC risk and survival analyzed in retrospective cohort studies has been not conclusive. Nevertheless, clinical trials are ongoing with so little information. So, this subject really needs to be more explored, so that more solid conclusions can be made. In order to make the novelty of our study more evident, we changed the text in the manuscript (lines 67-76 and 180-190).

Reviewer 3 Report (New Reviewer)

Comments and Suggestions for Authors

Manuscript titled “Aspirin increases the cytotoxic and colony-formation inhibitory 2 effect of metformin in PI3KCA-mutant colorectal cancer cells” by Gonçalves et all, is an in vitro study that exploring  effects of Aspirin and Metformin on colon cancer cells.

Papers has serious flaws:

Of Major importance:

·      Presented results are conflicting or not convincing. E.G:

1.     Figure 1 and 2: There are significant discrepancies present. In graph A, the Authors did not observe effect any on the proliferation of HT-29 cells when administered with 0.1 mM of metformin. In contrast, graph B shows a significant decrease in cell proliferation when HT-29 cells are exposed to 0.1 mM of metformin. This discrepancy is also evident in Figure 2.

2.     Figure 2, graphs E and F: The authors evaluated the effects of metformin on LDH production. However, major discrepancies exist between graphs E and F. In graph E, 0.1 mM of metformin reduces LDH activity, while in graph F, 0.1 mM of metformin induces LDH activity.

·      Figure 5: The authors should clarify why they did not observe any effect when administering 0.5 mM of aspirin alone, whereas the administration of 0.5 mM of aspirin in combination with metformin did have an effect In lines 284-285, the Authors assert that the concentrations of both Aspirin and Metformin proposed are "above the therapeutic blood levels of these drugs in humans." Consequently, the "repurposing strategy" that should ensue from these findings cannot be achieved.

·      Authors have improperly used the definitions of “cell proliferation”, and “cell viability”. This is a crucial point for this kind of proposed research.

·      But the most important problem of this manuscript is the lack of novelty: presented results are already published in literature by many Authors and well established.

·      Discussion is mostly speculative: it is not supported by obtained results

·      English needs to be revised……..

Minor revision:

·      Some of the proposed references are quite dated. Authors should be updated to ensure the relevance and currency of the literature cited.

·      In the introduction section, it is suggested to enhance the discussion about the differences between the two cell lines used in the study.

·      Lines 93-95 should be moved to materials and methods section since they do not express results.

·      In line 222, the authors should meticulously verify the names of the specified drugs.

·      Material and Method section must be completely rewritten since multiple redundancies are present (e.g.line 316).

Comments on the Quality of English Language

  English needs to be revised. 

Author Response

 Thank you for the revision. We next reply to your comments.

Of Major importance:

  • Presented results are conflicting or not convincing. E.G:
  1. Figure 1 and 2: There are significant discrepancies present. In graph A, the Authors did not observe effect any on the proliferation of HT-29 cells when administered with 0.1 mM of metformin. In contrast, graph B shows a significant decrease in cell proliferation when HT-29 cells are exposed to 0.1 mM of metformin. This discrepancy is also evident in Figure 2.

Answer: The results of the individual effects of the drugs and of their combination were obtained in distinct series of experiments.

In relation to Figure 1, there was indeed a discrepancy in the effect of metformin shown in Figures 1a and b. When we looked again at the results of metformin 0.1 mM presented in Figure 1a, we realized that we included all the experimental values obtained, but two of the values were in fact very odd. We decided, following your criticism, to remove them. A new figure 1a is now included in the manuscript. The effect of metformin is now similar in the 2 series of experiments.

In relation to Figure 2, there is no discrepancy. If you compare the effects of metformin 0.1 mM in a vs. b, c vs. d and e vs. f, the effects are quite similar. The same happens with aspirin 0.5 mM.

  1. Figure 2, graphs E and F: The authors evaluated the effects of metformin on LDH production. However, major discrepancies exist between graphs E and F. In graph E, 0.1 mM of metformin reduces LDH activity, while in graph F, 0.1 mM of metformin induces LDH activity.

Answer: When comparing graph 2e and 2f, the effect of metformin 0.1 mM is very similar. There is a tendency for an increase, which is however not significant. I do not understand your question. Perhaps you looked at metformin 0.01 mM (and not 0.1 mM) in graph 2e.

  • Figure 5: The authors should clarify why they did not observe any effect when administering 0.5 mM of aspirin alone, whereas the administration of 0.5 mM of aspirin in combination with metformin did have an effect.

Answer: we had an effect of aspirin alone on 14C-BT uptake, but no effect when aspirin and metformin were combined. Because metformin is able to affect other membrane transporters (eg. glucose transporters (Zhang et al. 2022)), it is possible that metformin acts at the transporter level (eg. by binding to the transporter) thus preventing aspirin-induced inhibition of BT uptake. But this is only a hypothesis, that we added to the manuscript (lines 257-261).

In lines 284-285, the Authors assert that the concentrations of both Aspirin and Metformin proposed are "above the therapeutic blood levels of these drugs in humans." Consequently, the "repurposing strategy" that should ensue from these findings cannot be achieved.

Answer: Yes, but there are strategies that may help solve this problem. We added a sentence referring this (lines 310-311).

  • Authors have improperly used the definitions of “cell proliferation”, and “cell viability”. This is a crucial point for this kind of proposed research.

Answer: we have measured cell proliferation rates by using the 3H-thymidine incorporation assay. This assay measures proliferation (µCi incorporation/mg cell protein). As to cell viability, we have used 3 distinct methods (LDH, resazurin and SRB). We have mentioned that in the manuscript (lines 377-381). I do not understand why the referee says that we improperly used the terms.

  • But the most important problem of this manuscript is the lack of novelty: presented results are already published in literature by many Authors and well established.

We cannot agree with you in that there is lack of novelty in this study. Only two previous studies tested the in vitro effect of the combination of metformin+aspirin on CRC cells. Of these, one used CRC cell lines distinct from the ones we used, and the other used patient-derived CRC cells. Moreover, these studies were rather limited as they only evaluated autophagy (one study) and tumor spheroid formation (the other). So, I do not understand why the referee says that there is a lack of novelty of this study. Moreover, the effects of this combination in relation to CRC risk and survival analyzed in retrospective cohort studies have been not conclusive, and clinical trials are nevertheless ongoing with so little and contradictory information. So, this subject really needs to be more explored, and more solid conclusions be made. In order to make the novelty of our study more evident, we changed the text in the manuscript (lines 67-76 and 180-190).

  • Discussion is mostly speculative: it is not supported by obtained results

Answer: in agreement with your criticism, we changed the manuscript (lines 29-31 and 317-319).

  • English needs to be revised…

Answer: the manuscript was read by an English expert.

Minor revision:

  • Some of the proposed references are quite dated. Authors should be updated to ensure the relevance and currency of the literature cited.

Answer: in agreement with your comment, older reviews were replaced by more recent ones. But as to original manuscripts, we have to keep them.

  • In the introduction section, it is suggested to enhance the discussion about the differences between the two cell lines used in the study.

Answer: done (lines 79-85).

  • Lines 93-95 should be moved to materials and methods section since they do not express results.

Answer: done (lines 377-381).

  • In line 222, the authors should meticulously verify the names of the specified drugs.

Answer: done (line 242).

  • Material and Method section must be completely rewritten since multiple redundancies are present (e.g. line 316).

Answer: redundancies in the Methods section were removed (lines 332-333, 342, 344, 372, 373).

Reviewer 4 Report (New Reviewer)

Comments and Suggestions for Authors

The authors have presented the work titled as "Aspirin increases the cytotoxic and colony-formation inhibitory 2 effect of metformin in PI3KCA-mutant colorectal cancer cells". The overall goal of the manuscript sounds interesting. There are minor issues which need to fixed.

1. The title needs to rewritten to avoid confusion.

2. I feel the finding of the authors fulfills the reality about the impact of aspirin and metformin mainly regarding PI3KCA. There are some works where PI3KCA appears to have such potential and thus in discussion/introduction these references could be cited ((a) https://www.mdpi.com/2073-4409/11/24/4121 and (2) https://www.mdpi.com/1422-0067/23/19/11024).

3. It would be worth and helpful to have respective citations in the method section and subsections.

Comments on the Quality of English Language

The quality of English is fine.

Author Response

Thank you for the revision. We next reply to your comments.

1. The title needs to rewritten to avoid confusion.

Answer: we changed the title. We hope that now it is less confuse.

  1. I feel the finding of the authors fulfills the reality about the impact of aspirin and metformin mainly regarding PI3KCA. There are some works where PI3KCA appears to have such potential and thus in discussion/introduction these references could be cited ((a) https://www.mdpi.com/2073-4409/11/24/4121 and (2) https://www.mdpi.com/1422-0067/23/19/11024).

Answer: in agreement with your comment, we added one of the citations you mention, as it refers specifically to CRC, in the Discussion (ref. 55).

  1. It would be worth and helpful to have respective citations in the method section and subsections.

Answer: in agreement with your suggestion, citations for SRB, 3H-thymidine incorporation, resazurin, LDH, wound healing and soft agar assays were added in the Methods (lines 369, 384, 397, 405, 418, 423).

Round 2

Reviewer 3 Report (New Reviewer)

Comments and Suggestions for Authors

xxx

This manuscript is a resubmission of an earlier submission. The following is a list of the peer review reports and author responses from that submission.

Round 1

Reviewer 1 Report

Comments and Suggestions for Authors

The paper by Joana Goncalves and colleagues is a well-written research article about the effect of the combination of metformin and aspirin in 2 different colorectal cancer cell lines.

I would suggest some minor revisions: 

The choice of the 2 cell lines is not completely clear to me. Actually, the 2 cell lines differ in BRAF and PIK3CA mutation status but they are similar in many aspects (MSS, CIN, p53-mutant, APC-mutant, KRAS-WT and PTEN-WT). Why did the authors choose only 2 cell lines and these 2 in particular? Since aspirin has been shown effective in CRCs with microsatellite instability, I would have added other cell lines with different characteristics.

In the Discussion, I would add a detailed paragraph about the limitations of the study (only 2 CRC cell lines, effects observed in vitro not completely concordant with in vivo studies, etc.).

Lines 36-41: This paragraph is too generic. I would delete it and go directly to the topic of the paper.

Reviewer 2 Report

Comments and Suggestions for Authors

The repositioning strategy of existing drugs has received a lot of interest in recent years. In the present manuscript, Gonçalves and colleagues evaluated in vitro cytotoxic effects of combination of aspirin and metformin on two colorectal cancer cell lines. There are many similar studies focusing on the effects of the combination of aspirin or metformin with other drugs, but these are not discussed in the present manuscript.

For example doi: 10.1016/j.intimp.2023.110350, doi: 10.3390/ijms21239017, doi: 10.1001/jama.2013.6599, doi: 10.18632/oncotarget.20972, doi: 10.1002/biof.1947, doi: 10.1002/jbt.22662, doi: 10.2147/OTT.S245091, doi: 10.3390/ijms19051478, doi: 10.1002/jcb.26898, doi: 10.3978/j.issn.2305-5839.2014.06.01, doi: 10.1186/s12967-023-04263-8, doi: 10.21873/anticanres.15410, doi: 10.1158/1940-6207.CAPR-13-0337.

The novelty and significance of this study is not clear. The introduction does not provide the necessary background information about the topic of the study.

It is easy to understand the results, but where is the originality (most of these effects are known)? The results are related to common experimental parameters like proliferation, viability, migration, colony formation, nutrient uptake, and without investigating some mechanisms or signaling pathways. The latter are only speculated on the basis of other studies.

However, it needs considerable improvements before considering for publication.

Reviewer 3 Report

Comments and Suggestions for Authors

This paper aims to investigate the effect of ASP and MET alone or in combination in relation to CRC risk in vitro on cell viability, proliferation, migration, colony-forming ability, and metabolic features of two CRC cell lines (HT-29 and Caco-2).

They also evaluated the effect of the MET and ASP combination in relation to nutrient cellular uptake and metabolism by HT-29 cells, by focusing on BT and glucose and found that their anticancer effects in HT-29 cells are not associated with a decrease in nutrient cellular uptake.

They also clearly stated the difference between HT-29 and Caco-2 cells which were isolated from human adenocarcinomas, however they possess varying grades of differentiation and different migratory and metastatic potential.

Based on their analysis, they reached at the reason of why the combination of these two drugs has different effect on the two types of cells: They suggested that inhibition of the PI3K pathway known molecular target of both MET and ASP is the common mechanism involved in the anti-CRC effect of both these drugs.

The manuscript is written comprehensively enough to be understandable; the authors addressed this aim by demonstrating the workflow of the designed study strategy starting talking by the Metformine and Aspirine medical properties and how their combination could reflect their antiproliferative effect on in vitro CRC cells.

The authors addressed their hypothesis and opinion in a reproducible way, they used enough number of analyses to prove their results. The paper stated the purpose, discussion and global implication are clearly stated and consistent with the rest of the manuscript; authors provided the required enough information in their discussion by using a good number of important articles talked about the subject.

The results were presented in a clear way which facilitate in reaching a conclusion based on their analysis to explain the reason of why the combination of these two drugs has different effect on the two types of cells: They suggested that inhibition of the PI3K pathway known molecular target of both MET and ASP is the common mechanism involved in the anti-CRC effect of both these drugs.

There are few points to follow up:

1-     Analysis: Authors did not state what is the positive/ negative control that they used in their test.

2-     In terms of novelty: The paper subject is like some previous papers in the literature, e.i: https://doi.org/10.1186/s12885-020-07564-z, that’s why the authors need to highlight what is the difference between their study and the previous studies in the literature, saying e.i: it is the first study of …..

3-     Most of the abbreviations were not explained at the first place they are mentioned.

4-     In vitro, in vivo, et al.: should be written in italic.

No plagiarism has been detected.

References: The authors followed the journal guidelines for some references.